**Data Availability Statement:** This dataset is a cleaned and supplemented version of the Power-Sharing Event Dataset (PSED) collected by Ottman and Vueller in 2014. Some variables have been

# The implementation of random survival forests in conflict management data: An examination of power sharing and third party mediation in post-conflict countries

**Andrew B. Whetten**[1,2]*, **John R. Stevens**[2], **Damon Cann**[3]

**1** Department of Mathematical Sciences, UW-Milwaukee, Milwaukee, WI, United States of America,
**2** Department of Mathematics and Statistics, Utah State University, Logan, UT, United States of America,
**3** Department of Political Science, Utah State University, Logan, UT, United States of America

* awhetten@uwm.edu

## Abstract

Time-to-event analysis is a common occurrence in political science. In recent years, there has been an increased usage of machine learning methods in quantitative political science research. This article advocates for the implementation of machine learning duration models to assist in a sound model selection process. We provide a brief tutorial introduction to the random survival forest (RSF) algorithm and contrast it to a popular predecessor, the Cox proportional hazards model, with emphasis on methodological utility for political science researchers. We implement both methods for simulated time-to-event data and the Power-Sharing Event Dataset (PSED) to assist researchers in evaluating the merits of machine learning duration models. We provide evidence of significantly higher survival probabilities for peace agreements with 3rd party mediated design and implementation. We also detect increased survival probabilities for peace agreements that incorporate territorial power-sharing and avoid multiple rebel party signatories. Further, the RSF, a previously under-used method for analyzing political science time-to event data, provides a novel approach for ranking of peace agreement criteria importance in predicting peace agreement duration. Our findings demonstrate a scenario exhibiting the interpretability and performance of RSF for political science time-to-event data. These findings justify the robust interpretability and competitive performance of the random survival forest algorithm in numerous circumstances, in addition to promoting a diverse, holistic model-selection process for time-to-event political science data.

## Introduction

In modern statistical methodology, two primary classes of predictive models exist. The traditional modeling approach assumes the data fits a stochastic model, whereas the algorithmic modeling approach, commonly referred to as machine learning, assumes no functional

described using their original codebook file. Third party mediation characteristics have been collected and supplemented to this dataset by Chong Chen in 2015. The raw data may be found at the following link: https://dataverse.harvard.edu/dataset.xhtml?persistentId=doi:10.7910/DVN/29657. Since the data is low-dimensional, we have provided our cleaned data as a csv file as a Supporting information file.

**Funding:** The author(s) received no specific funding for this work.

**Competing interests:** The authors have declared that no competing interests exist.

structure and instead seeks to uncover the structure of the data—the primary goal being the prediction onto the response variable [1]. The Cox proportional hazards model is a versatile alternative to parametric survival models in political science [2, 3]. The increased emphasis of predictive modeling and the recent introduction of tree-based methods in political science provokes further discussion of the extension of tree-based models to all branches of political science [4–8]. In this article, we demonstrate the suitability and flexibility that is gained from expanding time-to-event analysis to algorithmic models within the political science context by a comparison of the Cox proportional hazards and random survival forests (RSF) models [9].

We evaluate both methods using simulated data and the Power-Sharing Event Dataset in order to outline recommendations to researchers with time-to-event data [10]. For traditional methods, such as the Cox proportional hazards model, the nuances of residual diagnostics, variable selection, model regularization, and influential points must be addressed to meet model assumptions and optimize model performance. For many real-world data problems, this process involves several steps and involves some level of subjectivity. The implementation of the RSF requires minimal tuning to be considered optimal, and, as a result, random forest methods are considered exceptionally user-friendly methods.

In the simulated data section of this paper, we provide a scenario in which the baseline Cox model is uninterpretable. Subsequently, we construct a RSF for this example and demonstrate the interpretive advantages of the RSF.

In the following section, we analyze the Power-Sharing Event Dataset (PSED) and demonstrate an innovative approach to analyzing peace agreement data using the RSF. We document these beneficial insights with a brief discussion of each.

Although not a panacea, the use of the RSF approach demonstrates a number of advantages that are of important consideration for political science time-to-event data, including the following areas: (1) Higher dimensional data, (2) presence of collinearity, (3) violation of model assumptions, (4) the use of variable importance vs. statistical significance, and (5) detection of nonlinear effects. In terms of user experience, the RSF approach requires some time and attention in the construction of effective visualizations of the model results that account for the areas listed above, whereas with the Cox model more time is spent upfront diagnosing problems.

We discuss each of these items in more detail in the conclusion section.

## Materials and methods

### Random survival forests

Survival trees stem from the concept of regression trees, in which we recursively split the observations into groups, or nodes, that optimize the difference between groups using a metric that maximizes the difference in the response [11]. Survival trees, utilizing time-to-event as the response, primarily differ from regression trees in their ability to account for censored responses.

We provide an example of a survival tree in Fig 1 [12]. In this example, there are n = 150 observations, the response variable is time-to-event, and multiple covariates are available on each subject. At node 1, the algorithm splits the observations across the x2 covariate, which is a binary covariate (0,1). The resulting daughter nodes are each split again across the x11 covariate at 9.667 and 7.202 respectively. This process continues until the desired tree depth is created. In the resulting survival tree, six terminal nodes exist, and we provide their node sizes and their nonparametric survival curves for comparison. These survival curves visualize the effect of the different covariate values on the probability of survival beyond each time point.

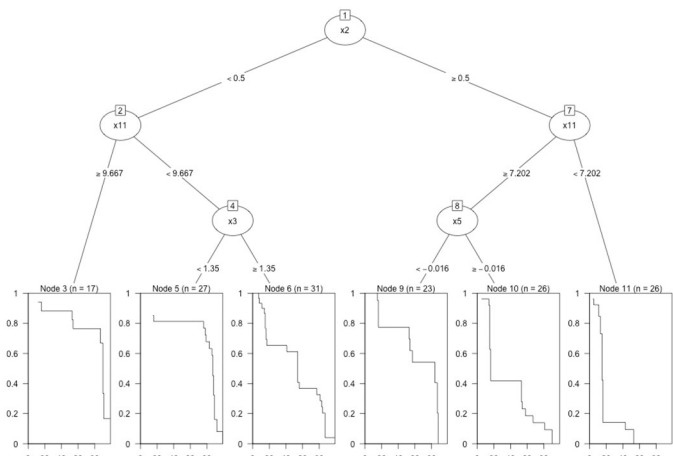

**Fig 1. Sample survival tree.** A sample survival tree with 6 terminal nodes. Survival trees are generally pruned to have larger terminal nodes to prevent over-fitting of the data. We provide estimated survival curves of each terminal node to assist in visualizing the difference between branches of the tree. Each node is represented by an oval with the corresponding covariate on which the node was subsequently split across. We observe that the partitioned data into the terminal nodes yields estimated survival curves with substantial differences in survival probability over the duration of the study.

The RSF algorithm is an ensemble machine learning method that constructs numerous survival trees from bootstrap samples of the data and utilizes the averaged predictions of each tree to construct an overall prediction of the survival time for each observation [13].

A common argument against implementing machine learning methods, such as random forest methods, is the "black box" construction of the model. The term "black box" stems from reduced understanding of internal structure of the model and predictor-response relationship. We emphasize that this is a misnomer for the RSF, and we outline the interpretative capabilities of the RSF using variable importance and partial dependence plots, which are visualized in the Results section. The results of a RSF, as with other machine learning methods, can indeed be interpretable, and the literature includes interpretabilty tutorials beyond those employed in this article [14].

Variable importance measures the net improvements to prediction error across all survival trees in the forest [9]. We compare the importance of all covariates in a variable importance plot. Variables with high, positive variable importance are identified to have stronger predictive relationships with the response. In contrast, variables with zero or negative variable importance identify weak predictors of survival. Variable importance is only quantifiable relative to other covariates in the model. This is because the accumulated improvements to model prediction achieved by a variable chosen at a node split are measured relative to the model's prior predictive performance.

Partial dependence plots represent the average random forest prediction of all variables excluding those of interest [13]. We construct partial dependence by fixing all other predictors at their average and examining the dependent effect of the covariate on the response at evenly spaced increments, and the predictions at each of these increments is plotted. In the context of time-to-event data, we examine the partial dependence at fixed time points through the study.

## The Cox model

Let $X_i = \{X_{i1}, \ldots, X_{ik}\}$ be the observed values for the $i^{th}$ subject. The well-known Cox model has the form $h_i(t) = h_0(t) \exp(\beta_1 x_{1i} + \beta_2 x_{1i} + \ldots + \beta_k x_{ki}) = h_0(t) \exp(X_i \beta)$, where $h_0(t)$ is the

baseline hazard function at time t and the terms comprising $X_i\beta$ are the covariates and their respective regression parameters. The hazard rate for an observation is assumed to be proportional to the baseline hazard function. The exceptional utility of the Cox model lies in the choice of an unparameterized baseline hazard function, which implies that no particular distribution is specified on the duration times. The frequent application and methodological research emerging from use of the Cox model in political science has stemmed from its preferable characteristics for event history modeling [2, 15–17].

In general practice, the Cox model provides competitive performance with increased flexibility over other parametric models that assume a distribution on the time-to-event response. Collinearity and a large number of predictors both produce issues with the Cox model that require variable selection using traditional stepwise selection techniques or model regularization using ridge, lasso, or elastic net [18]. We outline an alternative approach by means of machine learning methods, where these traditional steps to model selection are of lesser concern. More importantly, we seek to identify the types of questions that are explorable with the RSF algorithm.

## The Power-Sharing Event Dataset

The Power-Sharing Event Dataset (PSED) documents the power-sharing arrangements and civil war recurrence between government-rebel dyads from 1988 to 2006 [10]. The data comprises 79 distinct peace agreements during the first five years of implementation. All peace agreements with a duration exceeding the 5-year scope of the study are right censored. The PSED documents the qualitative features of all peace agreements as binary indicator variables. These include evaluations of the promise and implementation of economic, military, territorial, and political power-sharing, where (1, 0) denotes a (presence, absence) of a power sharing element. Binary covariates identifying the presence of 3rd party mediation during the design and implementation stage of the peace agreement process are included in the collection of covariates [19]. There is an inherent potential dependency structure for many observations in the PSED. As an example, 10 peace agreements in the PSED occurred in Chad. To address this issue, we construct a location variable with potentially dependent peace agreements labeled according to their respective countries, and a label of "Other" for peace agreements in countries where only one peace agreement was observed over the course of study. Such a variable allows retention of the vital dependence structures that may exist, and also a comparison between countries that observed one vs. multiple peace agreements.

The location variable is included as a predictor variable in both the Cox and RSF approaches. If peace agreements within a country tend to result in a dependence such that there is systematically higher or lower survival times within country, the inclusion of this variable will allow for the detection of such phenomenon. An interesting alternative to this approach would be to consider multi-state models [20], where within a country, the state of a peace agreement could alternate over time. A comparison of such an approach versus the method chosen here is worthy of future research. For the purposes of prediction of a new peace agreement in a given country, it would not be necessarily known if other peace agreements for this country would be forthcoming, and there is an inherent understanding that countries with multiple peace agreements are countries that have had terminate peace agreements. However, accounting for dependencies among observations by the inclusion of the location variable allows for a clarified understanding of the predictive effects of the other predictor variables which are of primary importance. Additionally, the issue of interdependence of peace agreements could altogether be avoided by only analyzing the time to the first recurrence of violence which would comprise 41 peace agreements here, but our efforts here are to

provide an analysis that comprehensively analyzes all peace agreements from this era and their respective attributes which may be adjusted for countries experiencing multiple agreements.

We emphasize that the objective in our construction of the location variable is to reduce induced dependence in our predictive analysis due to multiple observations within some countries. This dependence, if not accounted for, can disrupt the validity of the statistical model's inference. In the presence of such potential dependence, frailty models [21] can be used with the Cox regression model. Alternative tree-based survival analysis methods that specifically account for such clustered or correlated data may also be considered [22, 23].

## Results

### Simulations and RSF interpretation

We generate and analyze several simulated time-to-event datasets as a part of this project [24]. In order to keep this article focused on the general utility and interpretation of random survival forests, we do not focus on the technical details of our simulation here, but instead focus on general principles, with additional discussion in S2 File. We only include here one of our most instructive simulations, in which 50 predictors (49 random normal and 1 categorical) were constructed on a sample size of 150 observational units with arbitrarily induced collinearity. We simulate a time-to-event (or duration) response variable based on these predictors, where the duration variable is simply the linear combination of the predictor variable values multiplied by their respective effect sizes. To avoid complicating the assessment, we limit the magnitude of most predictor variables' effects to negligibly low effect sizes, and as such, we focus on detecting the predictive effects of only a few continuous variables and primarily the one categorical variable, all of which are determined by construction to be strongly related to duration. Inter-dependencies between predictors are induced by defining a 49x49 covariance matrix with entries of varying magnitudes, the purpose here being to induce collinearity rather than measure its magnitude [24]. The response variable range is defined from $t = [1, 100]$. Observational units that did not experience the event by the end of study are right censored. The censoring rate randomly imposed on the simulated data was set at 0.20. In this simulated example, the Cox model is uninterpretable in its current state; the model did not converge and as a result, its parameter estimates are not meaningful. The Cox approach in most applications will require further tuning, as demonstrated in our analysis of the PSED.

Although the RSF is not immune to the effect of collinearity in these circumstances, the RSF is more robust, and the model is interpreted for demonstrative purposes in this section. This does not imply that the RSF model is free of misinterpretation. The presence of highly collinear predictor variables must still be addressed, and, as opposed to the Cox model, the RSF model handles the challenges of interpretation of predictors in the visualization of the algorithm's results using minimal depth interaction plots and conditional partial dependence plots.

We present the variable importance plot for this simulation in Fig 2. In our model, "V4," "V20", and the strata variable, (x.str) are the strongest predictors of survival time. The strata variable was intentionally simulated to have a large effect size on time-to-event [25–28].

In Fig 3, we present the partial dependence plot (PDP) for variable V20 at $t = 50$. We identify increases in survival probability for observations as V20 increases in value. The predicted probabilities along the y-axis are not directly interpretable since the change in partial dependence is dependent on previously determined conditions outlined by the other predictors. Instead of using PDP's for direct interpretability, we inspect the plot for nonlinear predictor effects. There is evidence of a nonlinear, perhaps logistic, increase of survival probability as

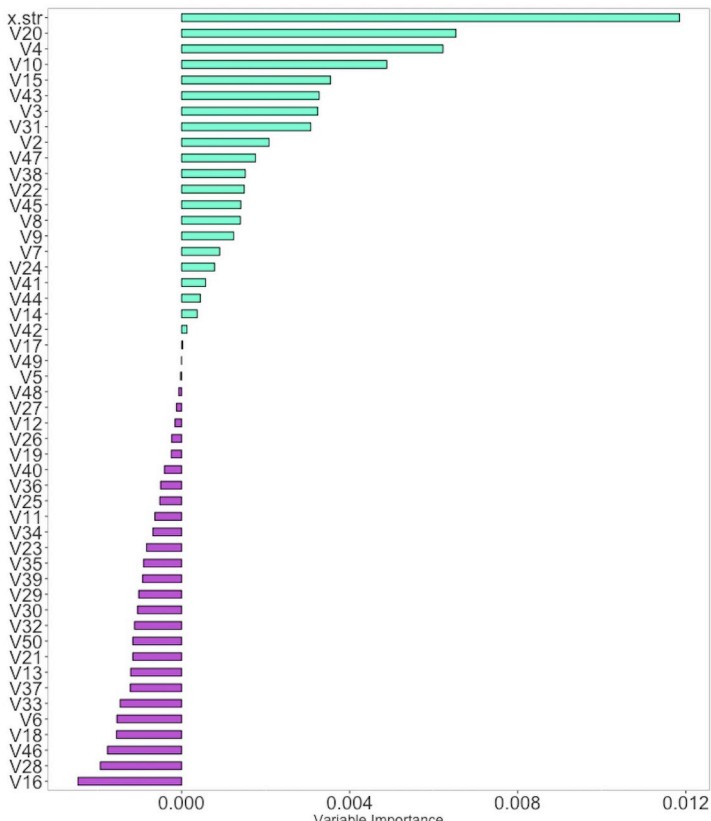

**Fig 2. Simulation variable importance.** Variable importance plot for all covariates in the RSF model. Substantial differences in variable importance measures distinguish variables that yield lesser improvements to prediction error from those that provide greater improvements.

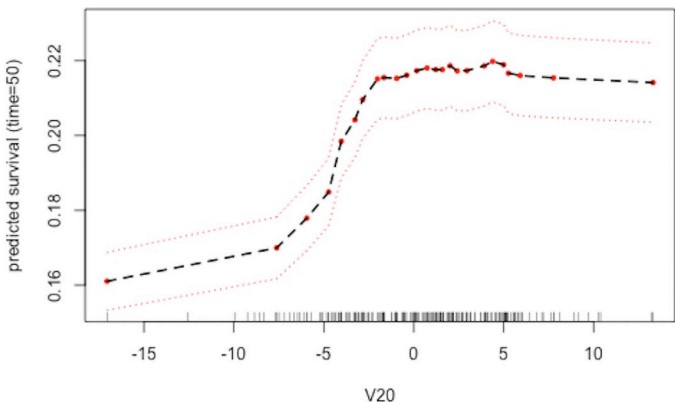

**Fig 3. Partial dependence plot from simulation data.** Plot shown is for the most important continuous variable in the model. Partial dependence can be inspected at any given time point in the model. The dark tick marks along the x-axis represent percentiles of the data with respect to the covariate of interest. The far boundary regions where the dark ticks are sparse identify regions where inferred relationships should be cautioned as few observations lie in these regions. Inferences outside the region (−10, 10) should be cautioned. Within this domain, the sharp increase in predicted survival probability as V20 increases followed by a stabilizing of survival probability characterize the relationship between V20 and survival.

V20 increases in value. V20 was one of the variables predetermined to have a noteworthy positive effect sizeon the response.

In these partial dependence plots, the confidence intervals are constructed using delete-d jackknife estimators [29]. In the presence of high correlations among predictor variables, the partial dependence plot may not be appropriate, as it implicitly assumes that one can vary the predictor of interest while holding fixed other predictors, even though they may be highly correlated with it. In such cases, alternative visualizations of effects would be advisable, such as accumulated local effects plots [14]. In addition, partial dependence plots are best suited for the visualizations of the effects of only one or two predictor variables at a time.

As demonstrated, variable importance and partial dependence provide an accessible approach interpreting random forest models while avoiding the need to detect and reduce collinearity present in the original selection of predictors.

## PSED analysis

In this analysis, we construct an RSF and regularized Cox model for the PSED. We compare their findings to demonstrate the interpretative advantages of the RSF approach using variable importance and partial dependence. In our analysis, we focus on detecting significant differences between the 4 mediation strata identified by the presence or absence of mediated design and implementation. We also explore the relationships between the most important predictors and peace agreement duration.

Variable importance measures can fail to identify relevant predictor variables in the presence strong correlations among variables [30]. We report the variance inflation factors of peace agreement characteristics in Table 1. For the PSED, there is minimal evidence of interfering correlatory structure. In cases where stronger dependence among predictor variables is detected, it would be advisable to consider an alternative approach such as recursive feature elimination [30] or conditional variable importance [31].

The RSF model shows evidence of marginally improved predictive performance over the baseline Cox model (RSF median concordance = 0.671, Q1 = 0.632, Q3 = 0.711; Cox median concordance = 0.663, Q1 = 0.604, Q3 = 0.725), which comprised of all variables in the original PSED. Predictive performance was performed using bootstrap sampled concordance

**Table 1. Variance inflation factors for peace agreements characteristics.**

| PA Characteristic | VIF |
|---|:---:|
| *Mediated Imp* | 4.761 |
| *Mediated Design* | 7.244 |
| *Design − Imp Interaction* | 10.867 |
| *Multiple Rebel Signatories* | 1.242 |
| *Conflict Duration* | 1.781 |
| *Conflict Intensity* | 4.595 |
| *Presence of UN Peacekeepers* | 1.361 |
| *Political Power Sharing* | 1.821 |
| *Military Power Sharing* | 1.399 |
| *Economic Power Sharing* | 1.554 |
| *Territorial Power Sharing* | 1.181 |
| *PPS Promise of Imp* | 2.738 |
| *MPS Promise of Imp* | 5.156 |
| *EPS Promise of Imp* | 1.504 |
| *TPS Promise of Imp* | 1.160 |

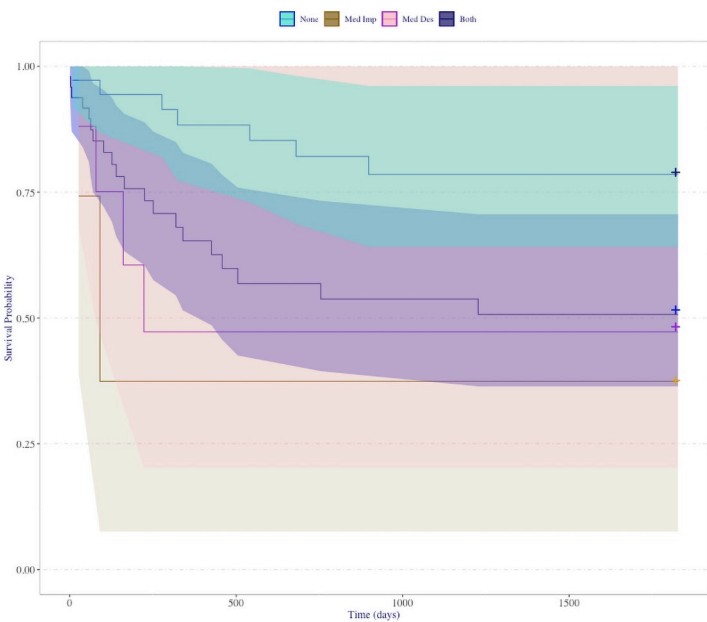

**Fig 4. Elastic net Cox survival curves.** Survival probabilities for elastic net selected Cox model. Coloring of curves and confidence bands corresponds to the presence of mediated design and implementation during the peace agreement process. The large spanning confidence bands for curves is expected since levels of third party mediation vs third party implementation of peace agreements has some sparse levels.

(C-index) error rates and the distributions of error rates were compared using 100 bootstrap samples [9]. (Besides the C-index, alternative measures of predictive performance can be considered, including Brier scores and the Schemper/Henderson measure [32]). Evidence of proportional hazards violations, saturation of insignificant predictors, and collinearity of predictors suggest the need for further model tuning. We regularize the Cox model using the elastic net penalties [18] to reduce the model dimensionality from 15 covariates to 6 covariates. The elastic-selected Cox model reported comparable predictive performance to the RSF.

In Fig 4, we report the survival curves for the elastic net selected Cox model. In Fig 5, we report the coefficient trace plot for the elastic net selected Cox model [33, 34]. We observe that some variables contain less stable parameter estimates and are ultimately dropped from the model. The optimizing 1-SE shrinkage parameter $s = .029$ leaves 6 covariates in the model. We note that the interaction of mediation design and implementation, drops quickly at first and then stabilizes. In Table 2, we provide the final table of parameter estimates for our Cox model at the 1SE shrinkage parameter location.

The interpretative issue present in this model arises from the sparse strata for the presence of only one of the 3rd party mediation criteria. The stratum with only the presence of mediated design comprises 7 peace agreements, and the stratum with the presence of mediated implementation comprises 3 peace agreements. The strata with no mediation and complete mediation had sample sizes of 40 and 29 respectively. Fig 4's wide confidence bands with exceptional overlap inhibit our ability to detect differences in strata survival probabilities. In the assessment of the differences between all mediation strata, we are left with inconclusive results.

Retrospectively, we examine the interaction of mediated design and implementation by grouping all peace agreements with complete mediation as done previously, and peace agreements with at least one or no mediation criteria in a second group. Visually, in Fig 4, this is the pooling of the lower three survival curves. In this simpler case, we successfully detect

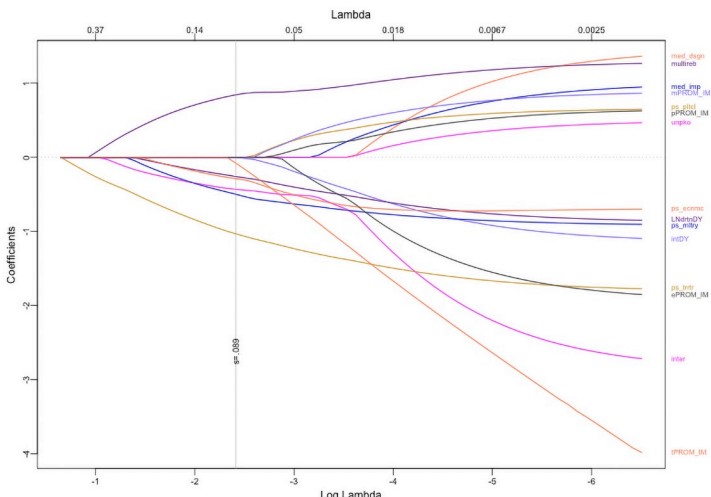

**Fig 5. Coefficient shrinkage plot.** Trace plot of elastic net Cox regression coefficients. The optimizing shrinkage parameter $s = 0.029$ is plotted as a vertical line marking the cutoff for the final model. The shrinkage parameter penalizes the regression model by restricting the net size of the magnitude of the regression coefficients. Increasing the penalization drives all regression coefficients to zero. Stability of variable relationships and presence of collinearity drives some regression coefficients to zero faster than others, and the vertical cutoff band at $s = 0.029$ identifies the penalized model with minimized 10-fold cross-validation error.

differences between the predicted survival probability of peace agreements that implement complete mediation and the remaining peace agreements (not shown).

Appropriately choosing, or tuning, parameters in a random survival forest helps avoid over-fitting [35], although such tuning can be computationally expensive for larger datasets. The reported RSF model was tuned using the randomForestSRC package in R [27], and the final selected model used 500 trees, 7 randomly selected variables for consideration at each node split, and an average terminal node size of 1. A general strength of random survival forest models is that minimal tuning is required. In Fig 6, the RSF model survival probability plot detects this result without the need for retrospective analysis. The RSF model in this case has marginally tighter confidence bands for the presence of both mediation criteria and absence of both criteria. As a result we gain the sufficient separation to conclude that there is a significant difference in survival probabilities between full and absent mediation. We note with great caution that the remaining sparse levels of this interaction also have alarmingly tight confidence bands. Although some evidence exists that having a single form of mediation has significantly lower survival probability than the presence of both forms, and likewise that having a single form of mediation does not yield significantly different survival probabilities than absent

**Table 2. Elastic net Cox regression coefficient estimates with the 1SE shrinkage parameter $s = 0.029$.**

| PA Characteristic | Estimate |
| --- | --- |
| *Design − Imp Interaction* | -0.153 |
| *MultipleRebelSignatories* | 0.350 |
| *ConflictDuration* | -3.096e-06 |
| *MilitaryPowerSharing* | -0.042 |
| *TerritorialPowerSharing* | -0.499 |
| *Location*: (*Other/Afghanistan*) | 0.190 |
| *Location*: (*Other/Somalia*) | 0.349 |

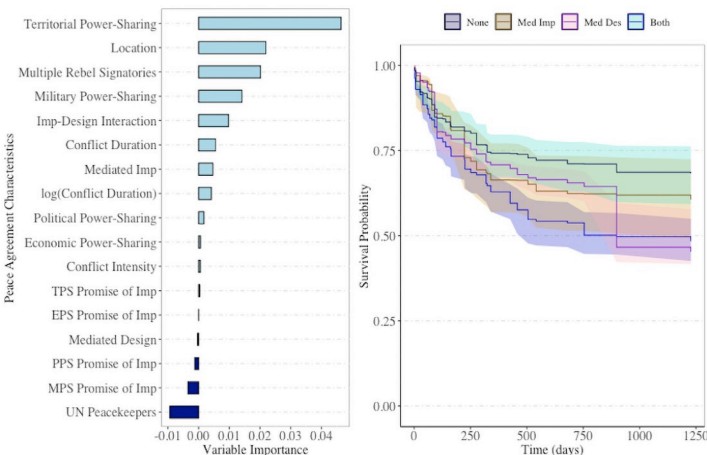

**Fig 6. RSF survival curves.** Right: Survival probabilities for RSF model. Coloring of curves and confidence bands corresponds to the presence of mediated design and implementation during the peace agreement process. Left: Variable importance plots for the PSED RSF model with 500 trees. Refer to S1 File for variable descriptions. Overlapping confidence bands are present, but they are notables tighter. This tightening of the confidence bands yields clear separation in survival probabilities between peace agreements with full third party mediation and those with none, a suspected relationship from Table 2.

mediation, we conclude that we have insufficient evidence to make this claim in this study, and it is of independent interest in future work to discuss the cause of tight confidence bands in the presence of sparse strata. Most importantly, we recall that the elastic model selection of the interaction variable indicates the significance of this peace agreement attribute, and the RSF model validates that a difference exists between at least one of the levels of our mediation variable.

In the left plot of Fig 6, we report the variable importance plot for the 15 covariates in the RSF model. In this plot, we identify territorial power-sharing, multiple rebel signatories, the interaction of mediated design and implementation, and military power-sharing as the most important criteria for predicting the survival of peace agreements. Note that all four of these variables remained in the lasso selected Cox model (Table 1).

In Fig 7, we examine the RSF partial dependence plots (PDPs) for territorial power-sharing, location, and multiple rebel signatories to study the predictive relationship of the covariates to peace agreement duration. We observe that both the presence of territorial power-sharing and the absence of multiple rebel signatories increase survival probability notably. This relationship is similar to the information from the elastic net Cox model parameter estimates in Table 1, which identified a significant decrease in the hazard of failure for an increase of 0 to 1 for territorial power-sharing (Estimate = -0.499) and an increase in the hazard for an increase of 0 to 1 for multiple rebel signatories (Estimate = 0.350). In the peace agreement location PDP, we identify that peace agreements occuring in a location with no other peace agreements during the course of study have the highest predicted survival probability. Afghanistan, Angola, Paupa-New Guinea, Sierra Leone, and Somalia are predicted to have peace agreements with the lowest survival probability.

We conclude that the presence of territorial power-sharing and the absence of multiple rebel signatories have a significant effect on long-term probability of survival of a peace agreement.

We emphasize that PDPs are best for examining the effect of only 1 to 2 variables at a time. Additionally, we have not yet addressed the effects of the dependency structure on the

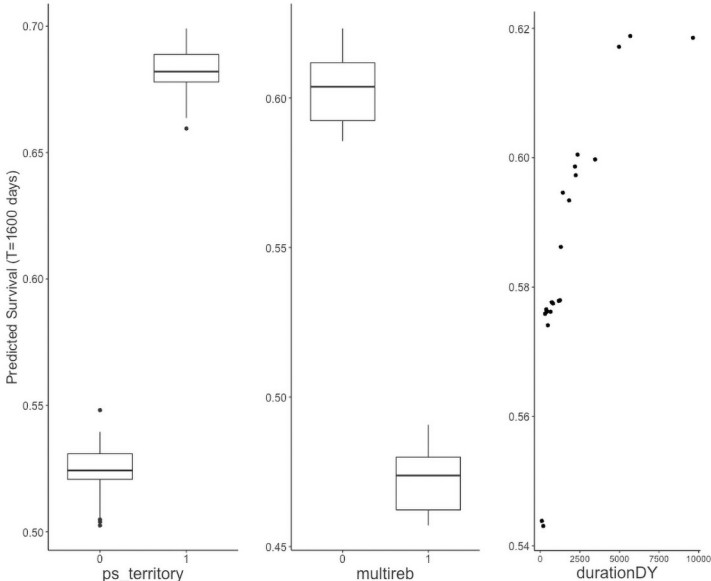

**Fig 7. Partial dependence plots for the PSED RSF model at 1600 days.** The y-axis denotes the survival probability observed at each level or value of a covariate for each observation. For categorical characteristics, the distribution of predicted survival probabilities is plotted using boxplots. Predicted survival is substantially higher for peace agreements that incorporate territorial power-sharing and avoid multiple rebel signatories. Peace agreements in "other" locations, referring to locations that did not have multiple peace agreements, are predicted to have higher survival probabilities.

predictor response relationships. In Fig 8 we report the pairwise interactions between covariates using interactive minimal depth [28]. In this plot, a higher interactive minimal depth recording denotes weaker interactive effects on peace agreement duration. We summarize a few important features of this plot. We present evidence of pairwise interactions between the presence of territorial power-sharing, multiple rebel signatories, conflict duration, and location.

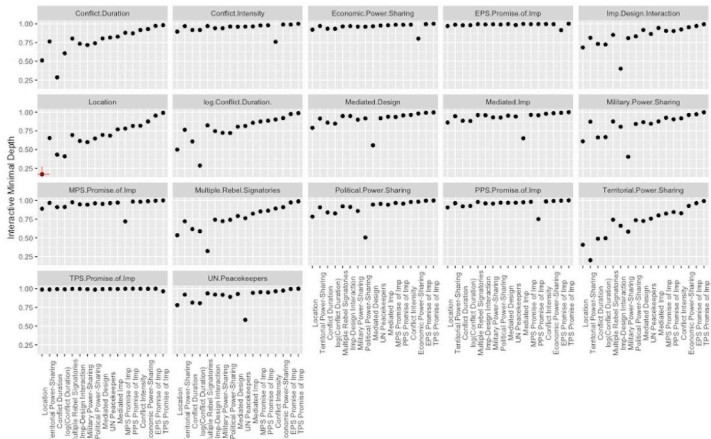

**Fig 8. Minimal depth interaction plots.** Interaction plots for all peace agreement characteristics and peace agreement location. Lower minimal depth values indicate stronger pairwise interactions between predictors.

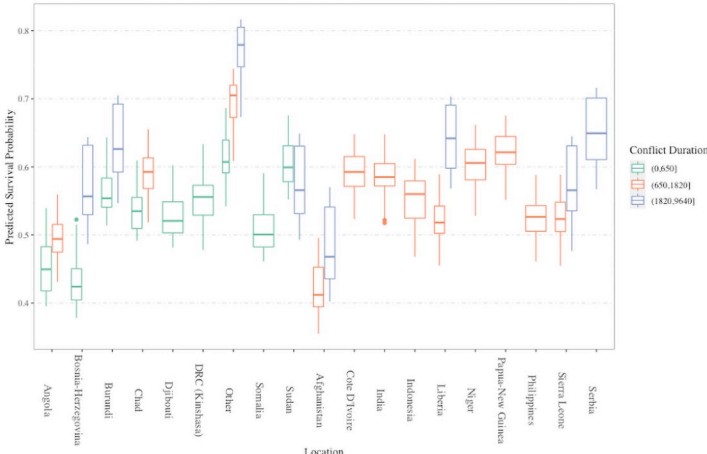

**Fig 9. Conditional partial dependence plot of location conditioned on conflict duration.** The effect of peace agreement location is conditioned on conflict duration using discretized grouping of conflict duration by quantiles [28]. We draw attention to the collection of boxplots corresponding to the "other" category. Within this group, substantial differences in survival probability are identified for various levels of prior conflict duration. Increasing conflict durations are associated with higher probability of survival of the agreement for sites with a single peace agreement over the duration of the study.

In Fig 9, we report the conditional partial dependence plot for location conditioned on conflict duration, to visualize the interaction of peace agreement location and conflict duration suggested by Fig 8. For each location in Fig 9, a series of boxplots of predicted survival probability are stratified by intervals of conflict duration. We chose intervals for previous conflict length discretization by evenly distributing the peace agreements in groups by quantiles. Locations that did not have conflict duration within a specific length did not have boxplots constructed. We identify that the effect of peace agreement location on peace agreement duration is dependent on conflict duration, where "short conflicts" (lasting less than about 650 days; see Fig 9) have lower survival probability than peace agreements at the same location with longer conflicts. We emphasize that due to the observational nature of the PSED, the validity and utility of these conclusions depend on the quality of the data and the representative nature of the peace agreements included in the PSED. While the PSED includes all known power-sharing arrangements and civil war recurrences between government-rebel dyads from 1988 to 2006 [10], these peace agreements must be considered representative of all possible peace agreements (in all relevant time windows) in order for conclusions from these data analysis methods (RSF or otherwise) to be meaningful.

## Discussion

In summary, we outline and discuss the unique discoveries found in the RSF analysis of the PSED that were not evident in the regularized Cox model. We emphasize that even though the RSF and elastic net Cox model did not have exceptional differences in computational or predictive performance, the RSF provides important interpretative advantages.

The RSF approach identifies a significant difference in predicted survival of peace agreements that involved third party mediation during the design and implementation of the agreement. We are limited in our ability to increase in the sparse levels of mediation, however, the elastic net Cox model alerts us to the importance of mediation as a factor in the model, and the pooling of at least one or no mediation strata into a single group confirms that the RSF result

are justified. We conclude from the RSF analysis that there is strong evidence that countries which incorporate both mediated design and implementation are predicted to have significantly higher duration of peace.

We can construct a ranking of peace agreement criteria by predictive importance (Fig 6). The presence/absence of territorial power-sharing in a peace agreement is the most important predictor of peace agreement duration. Of the available predictor variables, territorial power-sharing is twice as important as all predictors. Peace agreement location and the absence/presence of multiple rebel signatories are the second and third most important predictor in the model, and their importance is notably higher than the remaining peace agreement criteria. In the Cox model, we lack the ability to rank the predictive importance of the peace agreement criterion, and, as such, we fail to gain this insight using the regularized Cox model. Although an ad hoc notion of variable importance in the Cox model can be attained by examining the magnitude of effect sizes or the trace plot of each predictor, we emphasize here that these metrics are not guaranteed to rank in the same order as the RSF model. The RSF model ranks variables strictly based on their predictive value, and as such, variables with smaller effect sizes or lower significance may provide predictive information that may provide large improvements to predictive performance. The joint assessment of effect size and trace plot is prone to some level of subjectivity while the measure of variable importance, as used in the RSF model, eliminates this concern. In addition, the trace plot (such as in Fig 5) is best suited to show the stability of estimated coefficients rather than predictive performance of predictor variables. As a point of discussion, however, we do note that the six predictor variables retained in the elastic net Cox model (Table 2) correspond to the six most important variables identified by the RSF model (Fig 6).

One highly useful interpretative advantage possible from the RSF is the automatic consideration of interaction effects. The duration of prior dyadic conflict (durationDY) was identified to have an interactive relationship peace agreement location on predicted peace agreement duration (Fig 9) at a peace agreement survival time of 1200 days. As the prior conflict duration increases in length from short to medium conflict, the RSF model predicts noteworthy increases in the predicted peace agreement duration across several locations. This suggests that conflict duration still has an important effect on the survival probability of peace agreements in the latter years of its implementation, which is an insight unavailable from the elastic net Cox model and unavailable had we ignored the dependency structure in the PSED. Duration of prior dyadic conflict (durationDY) can be considered as providing some notion of the intensity or deep-rootedness of the conflict, which speaks to this variable's potential practical importance on the longevity of the subsequent peace agreement. Each peace agreement has its own value for this variable, being the time from beginning of the most recent armed conflict (which may have corresponded to the end of a previous peace agreement) until the end of the most recent conflict, even when multiple peace agreements are recorded within a single country. In our study, the potential dependence among multiple peace agreements within the same country is accounted for using our constructed "location" variable, and we emphasize that the exploration of alternative strategies to account for such dependence (such as by using multi-stage models [20]) is worthy of investigation in future work.

As outlined previously, we argue that assessing the relative importance of peace agreement characteristics is the most insightful way to prioritize negotiated criteria in the peace agreement process, and the RSF makes this approach possible in conflict management and peace science. Furthermore, we strongly encourage political scientists to consider the discipline-specific advantages gained from variable importance and partial dependence plots.

## Conclusion

The Cox proportional hazards model currently holds preference over other survival models in conflict management and most of the social sciences, and we emphasize the validity of this approach with the appropriate assessments of model diagnostics and assumptions. These extensions are competitive with the RSF approach and are appropriate considerations. In this paper, however, we emphasize that the RSF algorithm is the best "off-the-shelf" method for computational political scientists under less optimal conditions for traditional models, and the added interpretative benefits are vital to a modern and complete survival analysis.

In summary we outline common scenarios where an RSF approach should be considered:

- Higher dimensional data: As the number of covariates increases, the saturation of statistically insignificant covariates can inhibit effect size interpretation. The RSF approach requires no variable selection and retains robust interpretation of the predictors.

- Presence of Collinearity: Dependencies between predictors jeopardize model interpretability, and sound statistical practice requires model regularization or variable selection.

- Model assumptions are violated: Many parametric and semi-parametric models require the proportional hazards assumption to hold. The RSF model does not require this assumption, making the RSF the more convenient approach.

- Assessment of Predictive Importance vs. Statistical Significance: For some research questions, statistical significance may not be the most meaningful metric for identifying important covariates in the analysis. A highly significant predictor with a smaller hazard ratio estimate is subject to false positive conclusions. A marginally significant (or insignificant) predictor with a high hazard ratio estimate is subject to false negative conclusions. In either scenario, a degree of subjectivity is required. Variable importance is a highly interpretable approach to identifying the most important predictors of duration that measures the *predictive worth* of a covariate.

- Detection of nonlinear effects [36]: The utilization of partial dependence plots provides visualization of potentially complicated changes in the predicted response due to changes in one or more predictor variables. The disadvantage of this approach is that an optimized numerically quantifiable estimate of the relationship (i.e., a coefficient estimate or "slope") is not attainable as in Cox models, which assume a constant effect due to the (not always appropriate) proportional hazards assumption. However, when the true relationship is nonlinear, the RSF approach is able to detect (and visualize) it.

In light of the introduction of advanced machine learning methods to political science, we emphasize that strong statistical models should aim for improvements in predictive accuracy and preferably provide highly interpretable insight into the functional structure of the data. We encourage future work and creative usage of machine learning, such as RSFs, in political science.

## Supporting information

**S1 File. Description of variables.** PSED labeling of variables used in elastic net and rsf results section.
(CSV)

**S2 File. Supplemental discussion of simulation framework and findings.**
(PDF)

## Author Contributions

**Formal analysis:** Andrew B. Whetten.

**Methodology:** Andrew B. Whetten.

**Supervision:** John R. Stevens, Damon Cann.

**Validation:** Damon Cann.

**Visualization:** Andrew B. Whetten.

**Writing – original draft:** Andrew B. Whetten.

**Writing – review & editing:** Andrew B. Whetten, John R. Stevens, Damon Cann.

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
