## [Decision Letter · Decision Letter 0]

14 Sep 2020

PONE-D-20-22788

The implementation of the random survival forests in conflict management data: An examination of power sharing and third party mediation in post-conflict countries.

PLOS ONE

Dear Dr. Whetten,

Thank you for submitting your manuscript to PLOS ONE. After careful consideration, we feel that it has merit but does not fully meet PLOS ONE’s publication criteria as it currently stands. Therefore, we invite you to submit a revised version of the manuscript that addresses the points raised during the review process.

Reviewers find multiple issues that would need to be addressed, including tuning parameters and overfitting, correlation among predictors, and a lack of clarity in the interpretation of results.  A major reworking of the manuscript may be able to address these concerns.

We look forward to receiving your revised manuscript.

Kind regards,

Bryan C Daniels

Academic Editor

PLOS ONE

Journal Requirements:

2.We note that you have indicated that data from this study are available upon request. PLOS only allows data to be available upon request if there are legal or ethical restrictions on sharing data publicly. For more information on unacceptable data access restrictions, please see http://journals.plos.org/plosone/s/data-availability#loc-unacceptable-data-access-restrictions.

Reviewers' comments:

Reviewer's Responses to Questions

**Comments to the Author**

1. Is the manuscript technically sound, and do the data support the conclusions?

Reviewer #1: No

Reviewer #2: Yes

Reviewer #3: No

2. Has the statistical analysis been performed appropriately and rigorously? 

Reviewer #1: No

Reviewer #2: Yes

Reviewer #3: No

3. Have the authors made all data underlying the findings in their manuscript fully available?

Reviewer #1: Yes

Reviewer #2: Yes

Reviewer #3: Yes

4. Is the manuscript presented in an intelligible fashion and written in standard English?

Reviewer #1: Yes

Reviewer #2: Yes

Reviewer #3: Yes

5. Review Comments to the Author

Reviewer #1: In ``Conflict management data analysis using survival random forests'', Whetten et al. illustrate how to use Random Survival Forests (RSFs) to relax the parametric portion of Cox proportional hazards models in the context of an important question for political scientists and international relations scholars --- namely, the determinants of peace accord durability. While the paper is a welcome contribution to a body of work that illustrates the potential uses of various machine learning (ML) approaches in the study of domain-specific questions, the arguments advanced in favor of RSFs fall short on a few serious respects.

Perhaps the most serious concern raised by the manuscript is the authors' emphasis on the issue of predictor correlation and the use of variable importance measures to justify the use of RSFs. There is a well established literature demonstrating how various variable importance measures can fail to correctly identify relevant predictors in the presence of strong correlation among features (see, for instance, Gregorutti et al. 2017; or Strobl et al. 2008). In the context that Whetten et al. study, the power-sharing variables are likely to be correlated (although they show no evidence of this being the case). The authors make no mention of this issue, and they therefore run the risk of leading applied researchers in Political Science astray by this omission.

A second major issue is the lack of discussion of parameter tuning and the risk of overfitting in the context of RSFs. Like other RF approaches, RSFs require selection of tuning arguments, including the number of bagged samples, the proportion of features considered at each splitting node, and the tree-depth. On very large datasets, selecting the most appropriate values for these tuning arguments can be very computationally intensive. Moreover, care must be taken to respect the dependecy structures observed in data (such as those observed in the PSED data, which records multiple observations for the same government) when selecting parameters based on commonly used strategies, such as k-fold cross-validation. These are not ``minimal tuning'' tasks, and they deserve attention.

A couple of more minor issues compound these concerns. First, The authors state both that there is ``evidence of improved predictive performance'' of RSFs vs. the Cox model (p. 4, l. 138) and that ``the RSF and elastic net Cox model did not have exceptional differences in computational or predictive performance''. While these statements refer to the non-sparse and the sparse Cox models, respectively, it is an unnecessarily confusing set of statements, and makes me wonder what the measures of predictive performance used were (since the authors leave these unstated). Second, the authors never present the true data generating process behind their simulation (which is also unnecessarily vague in terms of the strength of induced correlations), which in turn makes it impossible to discern how well the variable importance measures capture the right set of predictive features in the simulation study. Third, there is no discussion of how confidence intervals are constructed for the RSF-based partial dependence plots; I just hope they are based on the infinitesimal jackknife (Wager et al. 2014), which is the most efficient, theoretically-guaranteed approach to building confidence intervals of nominal coverage in RFs. Finally (and this may be a pet-peeve) but there is no such thing as a ``highly significant'' result; under the NHST paradigm, a results is either significant or it is not.

Given these concerns, I am afraid I cannot recommend the current version of the manuscript for publication. However, I encourage Whetten et al. to address these issues, and offer applied researchers in Political Science a more decanted illustration of how ML models can be used to help questions of interest in the discipline.

Reviewer #2: The article provides a useful introduction to RSFs. The work that they present certainly highlight a number of benefits that could come from increases in the application of this approach. The sensitivity of canonical survival analyses is well understood in the literature even if no ideal solution has been agreed on. The alternative that the authors pose here certainly warrants increased attention in the literature.

However, a few questions/notes:

- The manuscript spends a notable amount of time highlighting how easy it is to implement an RSF, but more discussion is needed on how to train the model and avoid overfitting (e.g., number of trees, max features for splitting node, etc). Many packages in R and Python enable users to employ a random grid search over a set of hyperparameters, would that be the author's recommendation as well? This question should not be taken as me asking the authors to defend a certain strategy, but rather to highlight that when using these types of models scholars do need to actually have some sort of strategy.

- The utility of partial dependence plots in highlighting non-linear effects is certainly an advantage but in the context of high dimensional data, relying on PDPs can also be misleading. The maximum number of variables that we can typically examine via a PDP is two (and with a bit more work three). Say that we want to understand how quickly a intra-state conflict will end given the number of rebel groups and strength of the government's military. For the PDP of one of the features, e.g. # rebel groups, we assume that the other features (strength of government's military) is not correlated with the number of rebel groups, which in intrastate conflict research is a false assumption. For the computation of the PDP with a certain number of rebel groups, we average over the marginal distribution which might include a number of rebel groups that is unrealistic for a military of certain strength. Basically, meaning that when the features are correlated, we create new data points in areas of the feature distribution where the actual probability is very low. There are palliatives to this issue in the RF literature, but again what I think is most important is that readers of this work dont walk away thinking that RSFs are a panacea.

Reviewer #3: The core idea of this paper is to use a random forest variant of survival models as applied to a longstanding question in political science about the duration of accords.

There are a couple of things that really need to be improved here. First, the figures etc. appear to just be sort of default outputs from R with uninformative labels that are difficult to interpret.

Second, it is unclear what is new here at all. Neither partial dependence plots nor variable importance are novel to political science or other literatures. The papers the authors cite cover this same ground. If the aim is to make a methods contribution, it's hard to see what is being offered. To the extent there is a methods point being offered, it is that Random forests can be interpreted. But I don't see how that is true. There is no sense in wich these variable importance plots or patial dependence plots can be interpreted in a statistical sense? (Relatedly, where do those confidence intervals in the partial dependence plots even come from? What are their properties?) Remember that random forests are a greedy algorithm, not a true statistical model in a traditional sense. RF are a "black box" in the sense that the outputs have no clear statistically motivated meaning. If I am wrong here, then this should be spelled out in much greater detail and that would be a contribution.

To put a finer point on this, in the application the authors look at how the partial dependence plots find something significant while the Cox model does not. But the sample sizes for those groups are incredibly small, so we probably *want* huge confidence intervals. Perhaps the authors could show that RF has better coverage rates or better power/size visa simulation? But otherwise it seems like it is just overfitting.

Finally, much greater attention is needed to discussing tuning parameter selection.

6. PLOS authors have the option to publish the peer review history of their article (what does this mean?). If published, this will include your full peer review and any attached files.

Reviewer #1: No

Reviewer #2: No

Reviewer #3: No

---

## [Author Response · Author response to Decision Letter 0]

12 Oct 2020

Our response to reviewers is uploaded with our submission with the title "Response to Reviewers." We have addressed all items of reviewer feedback and have made the critical adjustments of tuning parameter optimization and accounting for the dependency structure in the data. We thank the reviewers for their feedback, and we look forward to the reviewers feedback on the significant adjustments to the manuscript.

---

## [Decision Letter · Decision Letter 1]

16 Dec 2020

PONE-D-20-22788R1

The implementation of the random survival forests in conflict management data: An examination of power sharing and third party mediation in post-conflict countries.

PLOS ONE

Dear Dr. Whetten,

Thank you for submitting your manuscript to PLOS ONE. After careful consideration, we feel that it has merit but does not fully meet PLOS ONE’s publication criteria as it currently stands. Therefore, we invite you to submit a revised version of the manuscript that addresses the points raised during the review process.

A reviewer has raised important issues about the interpretation of the model results, in particular questioning the interpretation of confidence intervals that was mentioned in the original reviews.  Further revisions may be able to address these issues.  Please note that the journal typically admits a maximum of two rounds of major revision.

We look forward to receiving your revised manuscript.

Kind regards,

Bryan C Daniels

Academic Editor

PLOS ONE

Reviewers' comments:

Reviewer's Responses to Questions

**Comments to the Author**

1. If the authors have adequately addressed your comments raised in a previous round of review and you feel that this manuscript is now acceptable for publication, you may indicate that here to bypass the “Comments to the Author” section, enter your conflict of interest statement in the “Confidential to Editor” section, and submit your "Accept" recommendation.

Reviewer #2: All comments have been addressed

Reviewer #4: (No Response)

2. Is the manuscript technically sound, and do the data support the conclusions?

Reviewer #2: Yes

Reviewer #4: No

3. Has the statistical analysis been performed appropriately and rigorously? 

Reviewer #2: Yes

Reviewer #4: No

4. Have the authors made all data underlying the findings in their manuscript fully available?

Reviewer #2: Yes

Reviewer #4: Yes

5. Is the manuscript presented in an intelligible fashion and written in standard English?

Reviewer #2: Yes

Reviewer #4: Yes

6. Review Comments to the Author

Reviewer #2: My main concerns revolved around the need for the authors to provide more discussion on tuning random forest models and drawing interpretations. The authors have provided a discussion of this issue that is minimal but satisfactory. Additionally, they have added in discussion about the limitations that come with PDPs, which is also satisfactory. The revisions the authors made have addressed the concerns that I highlighted.

Reviewer #4: General:

The paper provide a very brief tutorial and comparison of Random Survival Forests (RSF) models with the Cox proportional hazards model, including a so-called elastic net version of the Cox model. The comparison uses simulated data and a case study using the Power-Sharing Event Dataset (PSED). The authors claim that the RFS is the best "off-the-shelf" method for computational political scientists (under less optimal conditions), and that there is a added interpretative benefits compared to (e.g.) the elastic net version of the Cox model. In addition, they also claim to find significantly higher survival probabilities for certain peace agreements (for the PSED).

As a tutorial, the discussion is a bit short, and to get a complete introduction it is required to look up some of the references. This is not necessarily a problem, and the paper work well as a quick introduction to some of the main themes related to RFS. However, based on the discussion in the paper, some of the claims made about the RFS appear a bit too strong and unjustified. Also, the simulation study, and the corresponding discussion, is a bit short and perhaps too one-sided to provide a complete discussion. I understand that the main part of the simulation study is presented elsewhere, but it would have been nice with a more thorough discussion and perhaps also a more extensive literature overview to make this (as a tutorial) more complete and self-contained.

In addition, I think there are issues related to the case study that are not well discussed or explained in the paper:

1) Problems and consequences related to multiple observations/agreements within the same country.

2) And the introduction and use of the location variables and especially the "Other" variable in the model.

3) The surprisingly tight confidence intervals around the survival functions for the RFS.

These issues are discussed more below.

Abstract:

Is the ranking of peace agreement criteria importance really that novel? Is this because it has not been commonly used in this community before? Also, is the ranking very different from what would be the induced ranking from the elastic net Cox regression coefficient trace plots?

It is a bit unclear what is viewed as new regarding the justification for the "robust interpretability and competitive performance" for the RFS in this paper.

Introduction:

In my opinion, the discussion of the RFS is a little too sensational based on what is presented (as evidence) in this paper. This could be be solved by a more complete discussion and litteratur review; I assume there are several RFS studies/papers outside political science that could be of interest here.

Materials and methods:

* Random survival forests

* The Cox model

* The Power-Sharing Event Dataset:

The introduction and construction of the location and "Other" variables to correct for potentially dependent peace agreements appear suboptimal. It is also unclear how this is actually included in the model. In addition, the motivation for this correction and justification arguing that this will solve the problem is unclear?

Would a multi-state model be more suitable here?

Also, by introducing the variable "Other" in the model, are you not potentially including (part of) the response variable into the covariates? I understand that this does not have to be the case, but could this variable indirectly contain information about the (future) survival? Moreover, from the perspective of doing predictions (for e.g. a new case/country), this, that there will only be one peace agreement (if I understood the definition correctly), is not something that is known in advanced.

To avoid such problems and/or other problems related to dependency between observations for the same country, would it make more sense to study "time to first recurrence of violence" on the reduced dataset of 41 countries?

Results:

* Simulations and RSF Interpretation

Is it correct that the main finding in the simulation study is that "standard" estimation of the Cox model fail to converge when there is high level of multicollinearity? This is not very surprising, there are solutions to such problem, and, by itself, this is not a very convincing argument (although impractical) against using the Cox model. Are there other main insights from the simulation study that should also be included here?

In relation to Fig. 2, how does the induced ranking of the coefficient from the coefficient trace plot for the elastic net selected Cox model compare to the variable importance ranking? Is the ranking the same?

If the partial dependency plot in Fig 3. is based on the simulated data, how does the results correspond to the model used in the simulations? Is the conclusions/observations in accordance with how the simulation experiment was designed? This is not clear from the text.

* PSED Analysis

Is it possible to quantify this statement:

"The RSF model shows evidence of improved predictive performance over the baseline Cox model, which comprised of all variables in the original PSED."

I see that concordance is alluded to below, but there are no numeric values(?) to indicate if the difference is of any practical importance. Why not use additional measures, e.g. Brier score?

The confidence bands shown in Fig. 6 are surprisingly tight, especially when compared to Fig. 4 and what I obtain by a simple and quick re-analysis using the same data and standard models. In general, tighter confidence intervals is not evidence that a model is superior, these intervals could simply be wrong and/or have incorrect coverage probabilities (and interpretation); and in that sense be useless. Therefore, the following statement is questionable:

"The RSF model in many cases improves our ability to identify true differences between strata if they exist, especially with sparse strata."

And I do not see that there is any reference or discussion justifying that this is a valid and general statement.

Moreover, if this is actually the case, and the confidence bands have the correct interpretation, this is so surprising that it would be of independent interest to understand this in more details. For example, why is the RFS able to use the information so much more efficiently and/or what is wrong with the (assumptions) underlying the Cox model?

The conclusion at the end of page 7 feels a bit strong. I assume that this is not a truly randomised experiment. A more complete discussion regarding the validity of the conclusions focusing on the quality of data and assumptions underlying the analysis would be appreciated.

Discussion:

Is it possible to use the trace plots for the elastic net selected Cox model to interpret the importance of covariates? And, how and in what sense is the RFS providing an important interpretative advantage?

What is the practical importance of the "durationDY" variable? And, could the interpretation of this and other covariates be affect by having multiple observations from the same country in the same analysis?

7. PLOS authors have the option to publish the peer review history of their article (what does this mean?). If published, this will include your full peer review and any attached files.

Reviewer #2: No

Reviewer #4: No

---

## [Author Response · Author response to Decision Letter 1]

22 Jan 2021

Refer to "Response to Reviewers" File submission:

1/22/21 Response to reviews of manuscript "The implementation of the random survival forests in conflict management data: An examination of power sharing and third party mediation in post-conflict countries"

We thank the PLOS ONE Editors and reviewers for their time in considering this manuscript. After successfully resolving concerns of three previous reviewers, here we respond to feedback from a Reviewer #4. Our responses are in text boxes, and corresponding edits to the manuscript text appear in pink highlights of the manuscript PDF document. We specifically thank Reviewer #4 for their perspective and believe the resulting revisions have strengthened the manuscript, which we look forward to soon sharing with the readership of PLOS ONE.

Reviewer #4: General:

The paper provide a very brief tutorial and comparison of Random Survival Forests (RSF) models with the Cox proportional hazards model, including a so-called elastic net version of the Cox model. The comparison uses simulated data and a case study using the Power-Sharing Event Dataset (PSED). The authors claim that the RFS is the best "off-the-shelf" method for computational political scientists (under less optimal conditions), and that there is a added interpretative benefits compared to (e.g.) the elastic net version of the Cox model. In addition, they also claim to find significantly higher survival probabilities for certain peace agreements (for the PSED).

As a tutorial, the discussion is a bit short, and to get a complete introduction it is required to look up some of the references. This is not necessarily a problem, and the paper work well as a quick introduction to some of the main themes related to RFS. However, based on the discussion in the paper, some of the claims made about the RFS appear a bit too strong and unjustified. Also, the simulation study, and the corresponding discussion, is a bit short and perhaps too one-sided to provide a complete discussion. I understand that the main part of the simulation study is presented elsewhere, but it would have been nice with a more thorough discussion and perhaps also a more extensive literature overview to make this (as a tutorial) more complete and self-contained.

In addition, I think there are issues related to the case study that are not well discussed or explained in the paper:

1) Problems and consequences related to multiple observations/agreements within the same country.

2) And the introduction and use of the location variables and especially the "Other" variable in the model.

We have aimed for a balance here, focusing our discussion more on general principles, which we believe will be more useful for the intended audience rather than technical details. We have included additional discussion of the simulation and literature review, and hope the reviewer can appreciate the intended balance.

 3) The surprisingly tight confidence intervals around the survival functions for the RFS. These issues are discussed more below.

We have endeavored to respond to each of the concerns below, and thank the reviewer for their perspective.

Abstract:

Is the ranking of peace agreement criteria importance really that novel? Is this because it has not been commonly used in this community before? Also, is the ranking very different from what would be the induced ranking from the elastic net Cox regression coefficient trace plots?

 The novelty is in the use of RSF for ranking peace agreement criteria, not in the act of ranking. This clarification has been made in the abstract.

We thank the reviewer for highlighting their concern here, and a new paragraph, discussing the ranking of variables in the RSF procedure was added to the discussion section following the 3rd paragraph in an attempt to address this concern. In short, the primary purpose of trace plots to reflect stability of coefficient estimates, rather than assess the predictive importance of variables on the response. See lines 327-339

Ranking predictor variables is not as clear in a traditional regression-type model as it is in a Random Forest-type model. Is a larger effect size with a smaller standard error better than a moderate effect size with larger standard of error? Is it enough to sort on p-value? (This approach would fail in the presence of high collinearity among important predictors.) Does a larger effect size guarantee greater improvements to classification and MSE performance than a smaller effect size? (In our experience this is not the case as sometimes a variable that has lower significance and/or smaller effect size may still yield highly important improvements to performance in an RF model where the RF models finds that this variables improves predictions for some minority of observational units) In addition, we have seen examples of the subjectivity of the choice of variables in regularization models (like lasso and elastic net). While we do not believe this is worth emphasizing in the manuscript (where we instead refer simply to subjectivity concerns around line 334), for the purpose of addressing the reviewer concern, we can relate a generic example -- if two important predictors are collinear, their estimated coefficients can of course be unreliable, and it can happen that the lasso/elastic net model removes the less stable one and hence eliminates the strong collinearity between these two variables that was yielding alarming.

 It is a bit unclear what is viewed as new regarding the justification for the "robust interpretability and competitive performance" for the RFS in this paper.

 The novelty is in the application of RSF for such data in the field of political science. Our findings demonstrate a scenario exhibiting the interpretability and performance of RSF for such data. Our updated Abstract puts this novelty in better context.

 Introduction:

In my opinion, the discussion of the RFS is a little too sensational based on what is presented (as evidence) in this paper. This could be be solved by a more complete discussion and litteratur review; I assume there are several RFS studies/papers outside political science that could be of interest here.

Materials and methods:

* Random survival forests

* The Cox model

* The Power-Sharing Event Dataset:

The introduction and construction of the location and "Other" variables to correct for potentially dependent peace agreements appear suboptimal. It is also unclear how this is actually included in the model. In addition, the motivation for this correction and justification arguing that this will solve the problem is unclear?

Would a multi-state model be more suitable here?

Also, by introducing the variable "Other" in the model, are you not potentially including (part of) the response variable into the covariates? I understand that this does not have to be the case, but could this variable indirectly contain information about the (future) survival? Moreover, from the perspective of doing predictions (for e.g. a new case/country), this, that there will only be one peace agreement (if I understood the definition correctly), is not something that is known in advanced.

 We agree it is important to avoid over-sensationalizing any given method, including RSF. We have added additional mention of limitations / challenges in the use of the RSF. See lines 33-36 and 163-167.

 We thank the reviewer for alerting us to the needed details about the use and construction of this variable. We have made adjustments in the corresponding paragraph added at the end of the "The Power-Sharing Event Dataset" subsection. See lines 118-136.

 This is a fundamentally different research question, and one that we agree may be interesting to consider in a future work. We have mentioned this possible future research question in the paragraph added at the end of the "The Power-Sharing Event Dataset" subsection. See lines 121-124

 “Other” is just one possible level of the location variable, representing countries with only one peace agreement. The fact that these countries could tend to have systematically higher or lower response variable values is precisely the reason to include these countries in the same level of the location variable. We agree that for the purposes of prediction, for new peace agreements in a given country it would not necessarily be known if other peace agreements for the country would be forthcoming. However, accounting for dependence among observations (by the inclusion of the location variable) does allow for a clearer understanding of the predictive effects of the other predictor variables in the model, and that is the key.

We have made further adjustments to this section to bring light to this topic. See lines 125-131

 To avoid such problems and/or other problems related to dependency between observations for the same country, would it make more sense to study "time to first recurrence of violence" on the reduced dataset of 41 countries?

Results:

* Simulations and RSF Interpretation

Is it correct that the main finding in the simulation study is that "standard" estimation of the Cox model fail to converge when there is high level of multicollinearity? This is not very surprising, there are solutions to such problem, and, by itself, this is not a very convincing argument (although impractical) against using the Cox model. Are there other main insights from the simulation study that should also be included here?

 That would involve a different research question, and given the additional data for subsequent peace agreements in countries whose first post-1988 peace agreements failed, it would not account for the available data. Our efforts here are to provide an effective analysis that comprehensively analyzes peace agreements from this era. See lines 131-136

 The key is that RSF methods are excellent off the shelf methods and the challenge is to effectively visualize the off the self findings in such a way that collinearity is accounted for. We acknowledge that although we avoid the need to assess model diagnostics in this approach, there is still some challenging work to appropriately visualize the finding. We have made additional and clear mention of this topic. See lines 33-36 and 163-167

We have also added a supplemental file S2 that presents more discussion of the simulation framework and findings. We consider this supplemental because it is not central to the main focus of this paper. Even so, we do not dwell on technical details in the S2 File. We hope the reviewer can appreciate the balance to keep the manuscript's focus.

 In relation to Fig. 2, how does the induced ranking of the coefficient from the coefficient trace plot for the elastic net selected Cox model compare to the variable importance ranking? Is the ranking the same?

 Trace plots are best suited to show the stability of the estimated coefficient rather than any notion of predictive performance. More information on this topic as been added to the manuscript based on previous feedback from the reviewer (see lines 327-339). It is possible to try to define some notion of variable importance from the elastic results, but this prone to subjectivity, and wouldn’t necessarily be appropriate to compare directly to the traditional definition of variable importance used in RF methods. The RSF approach eliminates this subjectivity that would need to be used in the elastic net cox model.

 If the partial dependency plot in Fig 3. is based on the simulated data, how does the results correspond to the model used in the simulations? Is the conclusions/observations in accordance with how the simulation experiment was designed? This is not clear from the text.

We thank the reviewer for bringing attention to this. Lines 178-179 now mention this, without focusing on technical details.

* PSED Analysis

Is it possible to quantify this statement:

"The RSF model shows evidence of improved predictive performance over the baseline Cox model, which comprised of all variables in the original PSED."

I see that concordance is alluded to below, but there are no numeric values(?) to indicate if the difference is of any practical importance. Why not use additional measures, e.g. Brier score?

We have added a brief summary of the performance box plots (using the median, Q1 and Q3 concordance.

The confidence bands shown in Fig. 6 are surprisingly tight, especially when compared to Fig. 4 and what I obtain by a simple and quick re-analysis using the same data and standard models. In general, tighter confidence intervals is not evidence that a model is superior, these intervals could simply be wrong and/or have incorrect coverage probabilities (and interpretation); and in that sense be useless. Therefore, the following statement is questionable:

"The RSF model in many cases improves our ability to identify true differences between strata if they exist, especially with sparse strata."

And I do not see that there is any reference or discussion justifying that this is a valid and general statement.

Moreover, if this is actually the case, and the confidence bands have the correct interpretation, this is so surprising that it would be of independent interest to understand this in more details. For example, why is the RFS able to use the information so much more efficiently and/or what is wrong with the (assumptions) underlying the Cox model?

We greatly appreciate the reviewer catching our miscommunication in our argument here. A major modification to this section has been made to clarify the findings here clarifying the strengths and weaknesses of our model/method. See lines 244-257.

The conclusion at the end of page 7 feels a bit strong. I assume that this is not a truly randomized experiment. A more complete discussion regarding the validity of the conclusions focusing on the quality of data and assumptions underlying the analysis would be appreciated.

Discussion:

Is it possible to use the trace plots for the elastic net selected Cox model to interpret the importance of covariates? And, how and in what sense is the RFS providing an important interpretative advantage?

What is the practical importance of the "durationDY" variable? And, could the interpretation of this and other covariates be affect by having multiple observations from the same country in the same analysis?

We believe the reviewer is referring to the discussion about the interaction between peace agreement location and conflict duration, and we have tempered the conclusions there, with a comment about the observational nature of these data. See lines 298-304

We believe that our response to the earlier comment about the trace plots should apply here as well (see lines 327-339). As to the RSF providing an important interpretative advantage, we have added a brief example in the Discussion section regarding the RSF's ability to identify potential interaction effects, which leads to additional discussion in the remaining paragraph there. See line 340.

We have acknowledged other ways in which interdependence of peace agreements can be implemented, and we have provided further clarification on the practical importance of dyadic conflict duration. See lines 349-360.

We thank the reviewer for their thoughtful and thorough remarks, and we feel that it has improved the quality of our work.

---

## [Decision Letter · Decision Letter 2]

26 Mar 2021

PONE-D-20-22788R2

The implementation of the random survival forests in conflict management data: An examination of power sharing and third party mediation in post-conflict countries.

PLOS ONE

Dear Dr. Whetten,

Thank you for submitting your manuscript to PLOS ONE. After careful consideration, we feel that it has merit but does not fully meet PLOS ONE’s publication criteria as it currently stands. Therefore, we invite you to submit a revised version of the manuscript that addresses the points raised during the review process.

Reviewers are largely positive about the updated manuscript but find a few remaining issues that call into question the interpretation of the results.  A minor revision of the text is likely to resolve these issues.

We look forward to receiving your revised manuscript.

Kind regards,

Bryan C Daniels

Academic Editor

PLOS ONE

Journal Requirements:

Reviewers' comments:

Reviewer's Responses to Questions

**Comments to the Author**

1. If the authors have adequately addressed your comments raised in a previous round of review and you feel that this manuscript is now acceptable for publication, you may indicate that here to bypass the “Comments to the Author” section, enter your conflict of interest statement in the “Confidential to Editor” section, and submit your "Accept" recommendation.

Reviewer #5: (No Response)

Reviewer #6: All comments have been addressed

2. Is the manuscript technically sound, and do the data support the conclusions?

Reviewer #5: Yes

Reviewer #6: Yes

3. Has the statistical analysis been performed appropriately and rigorously? 

Reviewer #5: Yes

Reviewer #6: No

4. Have the authors made all data underlying the findings in their manuscript fully available?

Reviewer #5: Yes

Reviewer #6: Yes

5. Is the manuscript presented in an intelligible fashion and written in standard English?

Reviewer #5: Yes

Reviewer #6: Yes

6. Review Comments to the Author

Reviewer #5: I like the idea of introducing new methodologies in fields that are not fully aware of the latest available data analysis tools. This is why this article is interesting and pertinent even thought the data analysis example has limitations.

The multiple peace agreements in some countries potentially induce dependence among the observations. Adding a dummy variable for the country is reasonable. If the goal is to only study and compare the predictive performance of the models, then it might be sufficient. However, any inference (i.e. test of hypothesis, confidence intervals) are questionable because of that. The authors should clearly mention this. Methods for clustered data exist but they would be out of the scope of this paper. The authors should at least mention some of them. Frailty models are well-known and can be used with (semi-) parametric models like Cox. For trees, Fan et al. (2006) propose a way to handle such data; see also the R package MST for an implementation.

The right way to evaluate the predictive performance and compare the models is to use test data that are not used to fit the model. Since the sample size is so low (n=79), using cross-validation might be the only choice. The Brier score and C-index are the two most popular performance methods and should be used.

Reference

Fan, Juanjuan, et al. "Trees for correlated survival data by goodness of split, with applications to tooth prognosis." Journal of the American Statistical Association 101.475 (2006): 959-967.

Reviewer #6: I'm a new reviewer for this manuscript. I've read the responses the authors provided to the reviewer suggestions from earlier rounds. I've also read the manuscript carefully. The presentation appears thorough to me as do the responses to the comments from the other reviewers. I see no major issues to point out, so at this late stage (from my perspective) I will not provide any minor suggestions for the authors of the manuscript. I will provide one very small suggestion that the authors can choose to ignore if they disagree: generally I recommend very detailed figure captions. The authors' captions are pretty thorough but could stand a bit more detail. The table could use a more detailed caption as well. Again, this is a very very minor point. Otherwise, I'm satisfied with what I've seen in terms of the updated manuscript and the response to the earlier reviewers.

7. PLOS authors have the option to publish the peer review history of their article (what does this mean?). If published, this will include your full peer review and any attached files.

Reviewer #5: No

Reviewer #6: No

---

## [Author Response · Author response to Decision Letter 2]

12 Apr 2021

We thank the reviewers for their time and expertise in considering this manuscript. The text boxes in our Response to Reviewers pdf file include our responses to specific reviewer comments, and we look forward to bringing this work to the PLOS ONE readership.

---

## [Editor Report · Decision Letter 3]

19 Apr 2021

The implementation of random survival forests in conflict management data: An examination of power sharing and third party mediation in post-conflict countries.

PONE-D-20-22788R3

Dear Dr. Whetten,

We’re pleased to inform you that your manuscript has been judged scientifically suitable for publication and will be formally accepted for publication once it meets all outstanding technical requirements.

Kind regards,

Bryan C Daniels

Academic Editor

PLOS ONE
---

## [Editor Report · Acceptance letter]

22 Apr 2021

PONE-D-20-22788R3 

The implementation of random survival forests in conflict management data: An examination of power sharing and third party mediation in post-conflict countries 

Dear Dr. Whetten:

I'm pleased to inform you that your manuscript has been deemed suitable for publication in PLOS ONE. Congratulations! Your manuscript is now with our production department. 

Kind regards, 

on behalf of

Dr. Bryan C Daniels 

Academic Editor

PLOS ONE